



# An explanation for the different climate sensitivities of land and ocean surfaces based on the diurnal cycle

Axel Kleidon[1] and Maik Renner[1]

[1]Biospheric Theory and Modelling Group, Max-Planck-Institut für Biogeochemie, Jena, Germany

*Correspondence to:* Axel Kleidon (akleidon@bgc-jena.mpg.de)

**Abstract.** Observations and climate model simulations consistently show a higher climate sensitivity of land surfaces compared to ocean surfaces, with the cause for this difference being still unclear. Here we show that this difference in temperature sensitivity can be explained by the different means by which the diurnal variation in solar radiation is buffered. While ocean surfaces buffer the diurnal variations by heat storage changes below the surface, land surfaces buffer it mostly by heat storage changes above the surface in the lower atmosphere that are reflected in the diurnal growth of a convective boundary layer. Storage changes below the surface allow the ocean surface-atmosphere system to maintain turbulent fluxes over day and night, while the land surface-atmosphere system maintains turbulent fluxes only during the daytime hours when the surface is heated by absorption of solar radiation. This shorter duration of turbulent fluxes on land then results in a greater sensitivity of the land surface-atmosphere system to changes in the greenhouse forcing because nighttime temperatures are then shaped by radiative exchange only, which are more sensitive to changes in greenhouse forcing. We use a simple, analytic energy balance model of the surface-atmosphere system in which turbulent fluxes are constrained by the maximum power limit to estimate the effects of these different means to buffer the diurnal cycle on the resulting temperature sensitivities. The model predicts that land surfaces have a 50% greater climate sensitivity than ocean surfaces, and that the nighttime temperatures on land increase about twice as much as daytime temperatures because of the absence of turbulent fluxes at night. Both predictions compare very well with observations and CMIP 5 climate model simulations. Hence, the greater climate sensitivity of land surfaces can be explained by its buffering of diurnal variations of solar radiation in the lower atmosphere.

## 1 Introduction

It has long been reported that the sensitivity of near-surface air temperatures over land is greater than over ocean, with land surfaces warming about 50% more strongly than ocean surfaces (Huntingford and Cox, 2000; Sutton et al., 2007; Boer, 2011; Byrne and O'Gorman, 2013). This phenomenon has also be found in observations, with the ratio remaining surprisingly constant through time (Lambert and Chiang, 2007). Several explanations have been put forth to explain this robust feature, including the role of heat transport (Boer, 2011), a balancing effect of oceanic heat storage (Lambert and Chiang, 2007), changes in evapotranspiration (Sutton et al., 2007) and changes in relative humidity (Byrne and O'Gorman, 2013). Also, Joshi and Gregory (2008) showed that this effect depends on the nature of the forcing, and the ratio of land warming to ocean warming of about 1.5 holding only for changes in the greenhouse forcing.



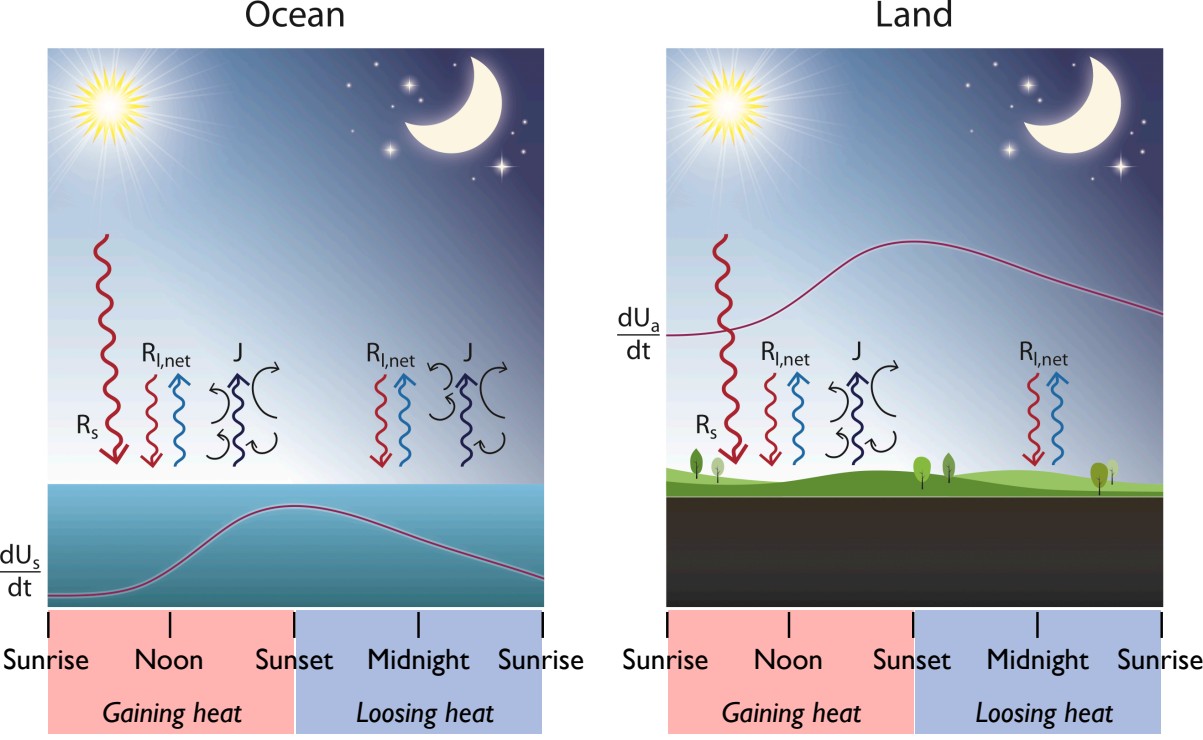

**Figure 1.** Schematic diagram of the surface energy balance of (a.) an ocean surface and (b.) the land surface. The main difference to explain the different temperature sensitivity is related here to the different way by which these surfaces buffer the diurnal variation of solar radiation. The ocean surface buffers it by heat storage changes below the surface in the upper ocean (shown by $dU_s/dt$) while the land surface buffers it primarily in the lower atmosphere (shown by $dU_a/dt$). Graphics: Annett Boerner.

Here, we explain this phenomenon of a higher climate sensitivity over land by the different ways of how the strong diurnal variation of solar radiation is buffered within the system (see Fig. 1). This buffering is accomplished by heat storage changes within the surface-atmosphere system that are forced by the heating by absorption of solar radiation during the day. The build-up of heat storage during the day then allows for nighttime temperatures that are far warmer than those one would expect in

5    the absence of solar radiative heating at night. For ocean surfaces, these heat storage changes take place in the surface ocean. Solar radiation penetrates the surface ocean to quite some depth before it is absorbed. Combined with the large heat capacity of water, this results in diurnal heat storage changes that take place below the ocean surface (sketched by the red line on the left of Fig. 1). The build-up of heat storage during the day then maintains radiative cooling and turbulent heat fluxes during the night, resulting in little diurnal variations in surface temperature and turbulent fluxes. Over land this situation is quite different.

10    Solar radiation is absorbed *at* the surface (or above in a canopy), but not below the surface. This is because land surfaces are not transparent as water, and because the heat conductivity of soils is generally so low that heat fluxes in and out of the ground are generally small. The strong diurnal variation in solar radiation is thus not buffered below, but rather above the surface in





the lower atmosphere. These changes in heat storage manifest themselves in the diurnal growth of the convective boundary layer. This buffering above the surface has an important consequence for the fluxes of the surface energy balance. Turbulent fluxes only take place when the surface is heated by solar radiation during the day that causes the near-surface air to become unstable, while the nighttime is characterised by stable conditions near the surface as little heat can be drawn from the heat

storage below the surface. This prevents turbulent fluxes to take place at night and the cooling at night is thus determined only by radiative exchange. Turbulent cooling of the surface thus takes place during half of the whole day, while the other half it is cooled by radiative exchange. This difference in cooling terms should make nighttime temperatures more sensitive to changes in the greenhouse effect than daytime temperatures, a well-known phenomenon reported in observations (Easterling et al., 1997), and this should result in a greater climate sensitivity of land surfaces as well.

We demonstrate this explanation with an extremely simple, yet physically-based energy balance model in which we incorporate the effects of where heat storage changes take place. The model yields analytic expressions for the different climate sensitivity of land and ocean surfaces as well as the different sensitivity of nighttime- and daytime temperatures. In the following, we first describe this model in section 2. We then illustrate the climatological mean state in section 3, derive analytic expressions for the ratios of these sensitivities, and compare these to CMIP 5 climate model simulations. Some of the limita-

tions of the model are then discussed, particularly regarding the description of terrestrial radiation and effects of the hydrologic cycle, and describe some of the implications. We close with a brief summary and conclusions.

## 2 Model description

Our model consists of the energy balances of the surface and the whole surface-atmosphere system, a parameterisation of terrestrial radiation that is based on the grey atmosphere approximation, a formulation of turbulent fluxes derived from the

thermodynamic constraint that these yield maximum power, and expressions for surface temperature that are derived from this model formulation. A schematic diagram of the model is provided in Fig. 2. The model formulation largely follows similar formulations used in previous studies (Kleidon and Renner, 2013a, b; Kleidon et al., 2014, 2015). The main modifications here relate to the representation of heat storage changes in the formulations and a formulation of terrestrial radiation based on the grey atmosphere approximation (as in Kleidon (2016)). The symbols used in the following description are summarised in Table

25 1.

### 2.1 Energy balances

For our description of the surface-atmosphere system, we need two basic energy balance constraints: the energy balance of the surface, and the energy balance of the whole system.

The surface energy balance is described in terms of heat storage changes that take place below the surface, $dU_s/dt$, the

absorbed solar radiation at the surface, $R_s$, the net cooling by longwave radiation, $R_{l,net}$, and the turbulent heat fluxes, $J$ (the sum of the sensible and latent heat flux, which are combined here for simplicity):

$$\frac{dU_s}{dt} = R_s - R_{l,net} - J \tag{1}$$



**Table 1.** Variables and parameters used in this study.

| Symbol | Variable | Units (or value) | Equation |
|---|---|---|---|
| $G$ | Convective power | W m$^{-2}$ | Eq. (8) |
| $J$ | Turbulent fluxes (of sensible and latent heat) | W m$^{-2}$ | – |
| $J_{opt}$ | Turbulent fluxes (optimized by max. power) | W m$^{-2}$ | Eq. (11) |
| $k_r$ | Radiative parameterization constant | W m$^{-2}$ K$^{-1}$ | Eq. (7) |
| $R_s$ | Surface absorption of solar radiation | W m$^{-2}$ | – |
| $R_{s,toa}$ | Total absorption of solar radiation (surface and atmosphere) | W m$^{-2}$ | – |
| $R_{l,net}$ | Net flux of longwave radiation at the surface | W m$^{-2}$ | Eq. (5) |
| $R_{l,net,opt}$ | Net flux of longwave radiation at the surface (optimized by max. power) | W m$^{-2}$ | Eq. (11) |
| $R_{l,0}$ | Radiative parameterization constant | W m$^{-2}$ | Eq. (6) |
| $U_s$ | Heat storage below the surface | J m$^{-2}$ | Eq. (1) |
| $U_a$ | Heat storage within the atmosphere | J m$^{-2}$ | - |
| $U_{tot}$ | Total heat storage | J m$^{-2}$ | Eq. (2) |
| $T_r$ | Radiative temperature | K | Eq. (3) |
| $T_s$ | Surface temperature | K | – |
| $T_{land}$ | Land surface temperature | K | Eq. (17) |
| $T_{ocean}$ | Ocean surface temperature | K | Eq. (14) |
| $T_{global}$ | Global mean surface temperature | K | Eq. (18) |
| $\phi$ | Ratio of land to ocean warming | - | Eq. (21) |
| $\tau$ | Longwave optical thickness | - | - |
| $\sigma$ | Stefan–Boltzmann constant | $5.67 \times 10^{-8}$ W m$^{-2}$K$^{-4}$ | – |

The energy balance of the whole column is described by:

$$\frac{dU_{tot}}{dt} = R_{s,toa} - \sigma T_r^4 \qquad (2)$$

where $dU_{tot}/dt$ represents the total change of heat storage within the surface-atmosphere system (consisting of heat storage changes below the surface, $dU_s/dt$, and within the atmosphere, $dU_a/dt$), $R_{s,toa}$ is the total absorption of solar radiation (surface and atmosphere) and $T_r$ is the radiative temperature by which radiation is emitted to space, and $\sigma$ is the Stefan-Boltzmann constant. The radiative temperature is determined from the mean energy balance taken over a sufficiently long time so that

$$T_r = \left( \frac{R_{s,toa,avg}}{\sigma} \right)^{1/4} \qquad (3)$$

where $R_{s,toa,avg}$ is the mean value of $R_{s,toa}$. We assume that $T_r$ does not change at the diurnal scale. This effectively represents our assumption that the system has sufficient capacity to store heat to balance out the variations in solar radiation.



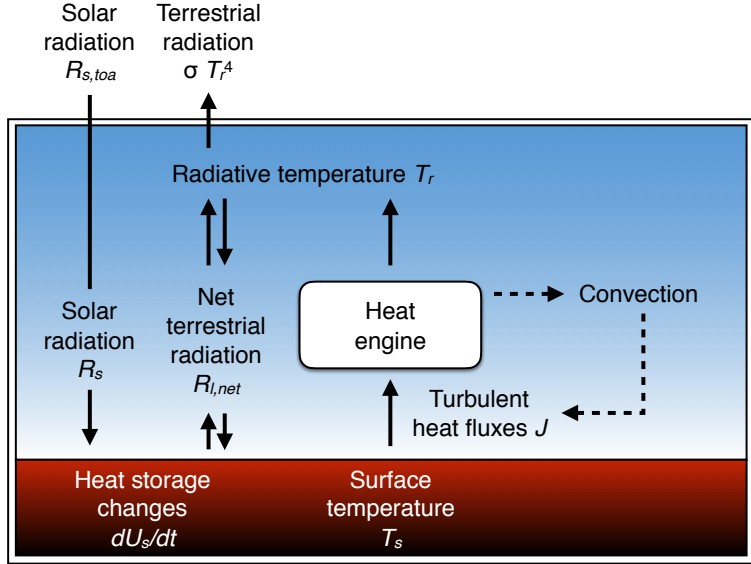

**Figure 2.** Schematic diagram of the surface energy balance used here in which turbulent heat fluxes are described as a result of an atmospheric heat engine operating between the surface and radiative temperatures. The limit to how much power can be derived from this heat engine provides a means to parameterise turbulent fluxes.

The total change in heat storage within the system can be determined from the assumption that this total heat storage does not change when averaged over the course of day and night. The total change in heat storage can then be inferred from the difference between the instantaneous and mean solar forcing, given by

$$\frac{dU_{tot}}{dt} = R_{s,toa} - R_{s,toa,avg} \tag{4}$$

5  In the following, we describe the ocean-atmosphere system as a system in which the heat storage changes take place below the surface (that is, in the surface ocean), so that $dU_s/dt = dU_{tot}/dt$. For the land-atmosphere system, we neglect the ground heat flux, which is typically small on a diurnal time scale, so that the change in heat storage needs to take place in the lower atmosphere to meet this diurnal energy balance constraint. As it turns out, this heat storage change does not enter the formulations explicitly so that the term $dU_a/dt$ does not appear in the equations.

10  ## 2.2  Parameterization of longwave radiation

To describe the net cooling by terrestrial radiation at the surface, $R_{l,net}$, we use a simple, linearised parameterisation of net longwave radiation of the form

$$R_{l,net} = R_{l,0} + k_r(T_s - T_r) \tag{5}$$

This formulation of net longwave radiation at the surface is similar to well-established empirical forms (e.g. Budyko, 1969),
15  but it can also be derived from the grey atmosphere approximation of radiative transfer in combination with a linearisation





of surface emission, as in Kleidon (2016). The latter has the advantage that the sensitivity of the parameters to a change in greenhouse forcing can directly be identified. In this derivation, the parameters $R_{l,0}$ and $k_r$ are related to the longwave optical depth, $\tau$, the emission of terrestrial radiation to space, $\sigma T_r^4 = R_{s,toa,avg}$, and the radiative temperature, $T_r$, with

$$R_{l,0} = \left(1 - \frac{3}{4}\tau\right) \cdot R_{s,toa,avg} \tag{6}$$

and

$$k_r = 4\sigma T_r^3 = 4 \cdot \frac{R_{s,toa,avg}}{T_r} \tag{7}$$

Note that a change in the greenhouse effect is associated with a change $\Delta\tau$, which alters the value of $R_{l,0}$, but not $k_r$. A change in absorption of solar radiation, for instance due to enhanced reflectance by clouds or aerosols, alters both expressions.

## 2.3 Turbulent fluxes determined from maximum power

The turbulent fluxes $J$ in the surface energy balance are derived from the assumption that these operate at the thermodynamic limit of maximum power (Kleidon and Renner, 2013a). In this formulation, turbulent fluxes are seen as the driver of a convective, atmospheric heat engine that generates the power to sustain motion and the turbulent exchange between the surface and the atmosphere. The power, $G$, or work per time, that this heat engine can provide is constrained by the Carnot limit, given by:

$$G = J \cdot \frac{T_s - T_r}{T_s} \tag{8}$$

where the driving temperature difference is set to the difference between the surface temperature and the radiative temperature. The motivation for using the radiative temperature as the cold temperature of the heat engine is to not use the temperature at a specific height of the atmosphere, but rather to the temperature at which the entropy export by radiative cooling to space is at a maximum. This temperature is, by definition, the radiative temperature, as it is the temperature of a blackbody that emits radiation of the amount $R_{s,toa,avg}$.

To derive the maximum power limit from the Carnot limit, we combine this limit with a fundamental tradeoff by which a greater turbulent heat flux results in a lower surface temperature, so that the derived power has a maximum power with an associated, optimum value of $J$ and $T_s$. This tradeoff is obtained by combining the surface energy balance (Eq. 1) and the expression for $R_{l,net}$ (Eq. 5) to express $T_s$,

$$T_s = T_r + \frac{R_s - dU_s/dt - R_{l,0} - J}{k_r} \tag{9}$$

When used in the expression for the Carnot limit (Eq. 8), we obtain

$$G = J \cdot \frac{R_s - dU_s/dt - R_{l,0} - J}{k_r T_s} \tag{10}$$

This expression has a maximum in power (i.e., maximum generation of turbulent kinetic energy), which can be derived analytically from $dG/dJ = 0$. When neglecting the variation of $T_s$ in the denominator, the maximisation yields an optimum heat



flux, $J_{opt}$, and net longwave flux, $R_{l,net,opt}$, given by:

$$J_{opt} = \frac{R_s - dU_s/dt - R_{l,0}}{2}$$

$$R_{l,net,opt} = \frac{R_s + dU_s/dt + R_{l,0}}{2} \tag{11}$$

Note how this formulation of surface energy balance partitioning depends on heat storage changes below the surface, $dU_s/dt$, but not on heat storage changes that take place in the lower atmosphere, $dU_a/dt$. We use these two contrasting cases of heat storage change to describe how this partitioning looks like for ocean (day and night) and land surfaces (daytime only). For simplicity, we assume that all of the absorbed solar radiation takes place at the surface (i.e., $R_s = R_{s,toa}$).

For the ocean surface, the dominant heat storage changes take place below the surface, so that $dU_s/dt \approx dU_{tot}/dt = R_{s,toa} - R_{s,toa,avg}$ (cf. Eq. 4). With this expression for $dU_s/dt$, the optimum surface energy partitioning is then given by:

$$J_{opt,ocean} = \frac{R_{s,avg} - R_{l,0}}{2}$$

$$R_{l,net,opt,ocean} = \frac{R_{s,avg} + R_{l,0}}{2} \tag{12}$$

This partitioning describes no temporal changes during the course of the day, as the turbulent fluxes as well as net longwave radiation are described by the mean solar radiation at the surface, $R_{s,avg}$, rather than the instantaneous forcing, $R_s$.

For the land surface, we assume that the heat storage changes take place in the lower atmosphere and $dU_s/dt \approx 0$. Then, the energy balance partitioning during the day is given by:

$$J_{opt,land} = \frac{R_s - R_{l,0}}{2}$$

$$R_{l,net,opt,land} = \frac{R_s + R_{l,0}}{2} \tag{13}$$

Note how this partitioning includes the instantaneous rate of absorption of solar radiation, $R_s$, thus resulting in a pronounced diurnal variation in surface energy balance partitioning as it is commonly observed on land. During night where $R_s = 0$ and $J = 0$ due to the prevalent stable conditions, we assume $R_{l,net} \approx 0$.

## 2.4 Surface temperatures

For the two contrasting cases of land and ocean surfaces, we can derive expressions for surface temperature by equating the optimum expressions for the net longwave radiative flux with eqn. 5.

For the ocean surface, surface energy balance partitioning does not change over the course of day and night. Hence, surface temperature is constant and depends only on the mean absorption of solar radiation:

$$T_{ocean} = T_r + \frac{R_{s,avg} - R_{l,0}}{2k_r} \tag{14}$$

For land, we split the surface energy balance partitioning into two parts of night and day. The nighttime temperature is derived directly from $R_{l,net} \approx 0$. This yields an equation for the nighttime temperature of

$$T_{night} = T_r - \frac{R_{l,0}}{k_r} \tag{15}$$





During the day, the mean absorption of solar radiation is about $R_{s,day} = 2R_{s,avg}$, which we use for $R_s$ in Eqs. 13. The mean daytime surface temperature is then given by

$$T_{day} = T_r + \frac{R_{s,day} - R_{l,0}}{2k_r} \tag{16}$$

This yields a mean surface temperature over land, $T_{land} = 1/2(T_{night} + T_{day})$, of

$$T_{land} = T_r + \frac{R_{s,avg} - 3/2 \cdot R_{l,0}}{2k_r} \tag{17}$$

When both temperatures are combined, the global mean surface temperature, $T_{global}$, is described by

$$T_{global} = (1 - f_{land})T_{ocean} + f_{land}T_{land} \tag{18}$$

where $f_l = 0.29$ is the fraction of land area of the total surface area of the Earth.

Eqs. 14 - 18 represent the key equations used in the following to evaluate the sensitivity of surface temperature to a change
in radiative forcing. These estimates are then compared to the respective sensitivities derived from the CMIP 5 climate model simulations (Taylor et al., 2012), using the 4xCO$_2$ and preindustrial control simulations (a list of models used is provided in Table A1).

## 3   Results and Discussion

### 3.1   Global energy balance

We first evaluate the energy balance partitioning and expressions for temperatures using global means. The forcing is described by the total mean absorption of solar radiation of the surface-atmosphere system of about $R_{s,toa,avg} = 240$ W m$^{-2}$, the mean absorption at the surface of $R_{s,avg} = 165$ W m$^{-2}$ (Stephens et al., 2012), and a longwave optical depth of about $\tau = 1.74$, which yields a mean surface temperature of about 288 K. The resulting values of the parameters $R_{l,0}$ and $k_r$, the surface energy balance partitioning, and the resulting values for the various temperature expressions are listed in Table 2. The turbulent
fluxes of 119 W m$^{-2}$ and net longwave radiation of 46 W m$^{-2}$ derived from the maximum power limit compare reasonably well to the estimates from observations of 112 W m$^{-2}$ and 53 W m$^{-2}$ (Stephens et al., 2012). Note that the radiative properties as well as continental area show strong geographic variations that are not accounted for here, so that this evaluation merely shows the plausibility of the formulations.

### 3.2   Temperature sensitivity to greenhouse forcing

We next evaluate the case of global warming. An increase in the greenhouse effect is represented in our formulation by an increase in the longwave optical depth, $\Delta\tau > 0$. We used a value of $\Delta\tau = 0.11$ to obtain a global temperature increase of $\Delta T_{glob} = 5.0$K. The increase in optical thickness then changes $\Delta R_{l,0} = -3/4 \cdot R_{s,toa} \cdot \Delta\tau < 0$.

The warming of the ocean surface, $\Delta T_{s,ocean}$, is then given by:

$$\Delta T_{ocean} = -\frac{\Delta R_{l,0}}{2k_r} \tag{19}$$



**Table 2.** Estimates for the global mean forcing, a global warming for a 4xCO$_2$ scenario, and a scenario of solar dimming.

| Symbol | Present day | Global Warming | Difference | Solar Dimming | Difference |
|---|---|---|---|---|---|
| **Forcing:** | | | | | |
| $R_{s,toa,avg}$ (W m$^{-2}$) | 240 | 240 | 0 | 240 | 0 |
| $R_{s,avg}$ (W m$^{-2}$) | 165 | 165 | 0 | 155 | -10 |
| $\tau$ (–) | 1.74 | 1.924 | 0.11 | 1.74 | 0 |
| **Derived radiation properties:** | | | | | |
| $T_r$ (K) | 255 | 255 | 0 | 255 | 0 |
| $R_{l,0}$ (W m$^{-2}$) | -73.2 | -106.3 | -33.1 | -73.2 | 0 |
| $k_r$ (W m$^{-2}$ K$^{-1}$) | 3.76 | 3.76 | 0 | 3.76 | 0 |
| **Predicted surface energy balance:** | | | | | |
| $J_{opt}$ (W m$^{-2}$) | 119 | 136 | 17 | 114 | -5 |
| $R_{l,net,opt}$ (W m$^{-2}$) | 46 | 29 | -17 | 41 | -5 |
| **Predicted temperatures:** | | | | | |
| $T_{ocean}$ (K) | 286.7 | 291.1 | 4.4 | 285.4 | -1.3 |
| $T_{land}$ (K) | 291.6 | 298.2 | 6.6 | 290.2 | -1.3 |
| $T_{global}$ (K) | 288.1 | 293.2 | 5.0 | 286.8 | -1.3 |

When the same change of optical thickness is applied to land, it results in a warming of the land surface, $\Delta T_{s,land}$, of:

$$\Delta T_{land} = -\frac{3}{2}\frac{\Delta R_{l,0}}{2k_r} \tag{20}$$

When taking the ratio, $\phi$, of these two changes, we obtain

$$\phi = \frac{\Delta T_{land}}{\Delta T_{ocean}} = \frac{3}{2} \tag{21}$$

5   Hence, the land surface is 50% more sensitive to a change in longwave optical depth than the ocean surface. We can also translate this fixed ratio between land and ocean warming into respective expressions that relate land and ocean warming to the global temperature change:

$$\Delta T_{ocean} = \frac{1}{1+0.5 f_{land}} \cdot \Delta T_{global} \approx 0.87 \cdot \Delta T_{global}$$
$$\Delta T_{land} = \frac{3}{2-f_{land}} \cdot \Delta T_{global} \approx 1.31 \cdot \Delta T_{global} \tag{22}$$

10  When using the global mean values as shown in Table 2, a global mean temperature increase of 5 K due to an increased greenhouse effect translates into an increase by 6.6 K over land (or 31% more than the global mean increase), while oceans only increase by 4.4 K (or 13% less than the global mean increase). In Fig. 3 we compare the predicted ratio $\phi$ as well as the temperature differences $\Delta T_{land}$ and $\Delta T_{ocean}$ to $\Delta T_{global}$ to the respective simulated values of CMIP 5 climate model





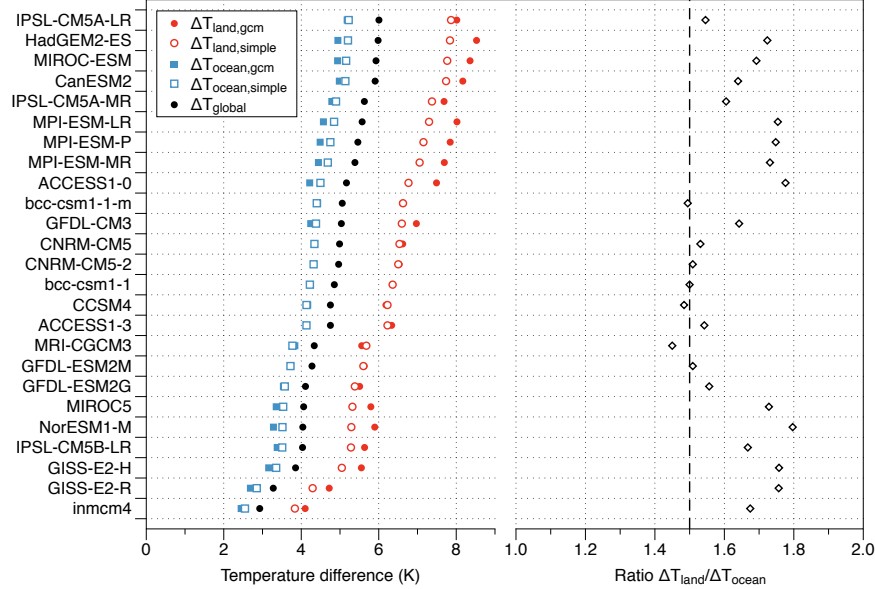

**Figure 3.** Global mean response of 25 CMIP 5 climate model simulations. Shown is the mean warming of the ocean surface ($\Delta T_{ocean,gcm}$, blue solid squares), the land surface ($\Delta T_{land,gcm}$, red solid circles), and the global mean ($\Delta T_{global}$, black dots) between the 4xCO2 and Preindustrial control simulations. Also shown are the equivalent changes ($\Delta T_{ocean,simple}$, blue open squares, and $\Delta T_{land,simple}$, red open circles) predicted from the energy balance considerations made here. The right diagram directly compares the ratio $\phi = \Delta T_{land}/\Delta T_{ocean}$ from the GCM simulations (diamonds) to the prediction made here (dashed line).

simulations. Although some deviations can be seen, our estimates overall compare very well to the global mean changes found in the CMIP 5 simulations.

As argued in the introduction, the difference in the climate sensitivity of land and ocean surfaces should be attributable to the different behaviour of the land surface at night than during the day. To evaluate this in our formulations, we also looked
5  at the sensitivities of nighttime and daytime temperatures as proxies for minimum and maximum temperatures. The minimum temperature typically occurs at the end of the night, and we approximate it by the use of $T_{night}$. The maximum temperature occurs at the end of the day, and for this temperature we use $T_{day}$. Using the above expressions for these temperatures, we obtain

$$
\begin{aligned}
\Delta T_{night} &= -\frac{\Delta R_{l,0}}{k_r} \\
10 \quad \Delta T_{day} &= -\frac{\Delta R_{l,0}}{2k_r} = \frac{1}{2}\Delta T_{night}
\end{aligned}
\tag{23}
$$

Hence, minimum temperatures increase about twice the rate than maximum temperatures in our formulation, thus reducing the diurnal temperature range. This is broadly consistent with observations, for which a range of about 1.6 - 2.4 is reported for most seasons (Horton, 1995; Easterling et al., 1997), although in observations the ratio varies between hemispheres and seasons.





We did not perform this evaluation for the CMIP 5 simulations for a few reasons. There are other effects, e.g, due to changes in the hydrologic cycle, as well as model biases that quite substantially affect the trend in the diurnal temperature range in the CMIP5 simulations (Lindvall and Svensson, 2015) so that this direct effect of an enhanced greenhouse forcing is not the dominant factor in the 4xCO2 simulations. These effects would need to be accounted for in our expressions before a more

detailed comparison could be made. Yet, the well-established observation that the diurnal temperature range decreases with global warming is consistent with our interpretation why the climate sensitivity of land is higher than for ocean surfaces.

### 3.3 Temperature sensitivity to solar forcing

To illustrate that changes in solar radiation affect the temperature sensitivity quite differently, as described in (Joshi and Gregory, 2008), we consider a case in which the total absorption of solar radiation is unchanged, but less solar radiation is absorbed

at the surface (i.e., $\Delta R_{s,toa,avg} = 0$, with $\Delta R_{s,avg} = -10$ W m$^{-2}$). This magnitude of solar dimming is comparable to the observed changes in absorbed solar radiation over the last decades (e.g. Wild, 2009), which in turn relates mostly to changes in aerosol concentrations in the atmosphere. For comparability, we use the same value for the longwave optical depth. The sensitivity to absorbed solar radiation at the surface is also evaluated in Table 2 in the column labeled "Solar Dimming".

Our simple estimates partition the reduction in surface absorption equally into reductions in $\Delta J$ and $\Delta R_{l,net}$ (cf. Eqs. 12

and 13). The change in ocean temperature is given by (cf. Eq. 14)

$$\Delta T_{ocean} = \frac{\Delta R_s}{2k_r} = \Delta T_{land} \tag{24}$$

and the land surface cools on average by the same amount (cf. Eq. 17). This is quite different than the result from the change in the greenhouse effect, where the sensitivities were different. The effect on the diurnal temperature range on land is also markedly different. While the nighttime temperatures remain unchanged as they do not depend on $R_s$, the daytime tempera-

tures are reduced by twice the mean cooling, with $\Delta T_{day} = \Delta R_s/k_r = 2\Delta T_{land}$. This effect of solar radiation on maximum temperatures is well known (e.g. Wild, 2009) and has been used to infer solar radiation from the diurnal temperature range (e.g. Bristow and Campbell, 1984). This equal reduction of ocean and land temperatures is also shown in the example estimate in Table 2.

Our formulations thus show that the temperature sensitivities of ocean and land surfaces, the sensitivities of minimum and

maximum temperatures on land, and thus of the diurnal temperature range are closely connected and react differently depending on the type of radiative change at the surface.

### 3.4 Discussion

Despite its physical basis, our model has, obviously, several potential limitations due to its extremely simple nature. These potential limitations relate to the parameterisation of radiation and turbulent fluxes, and on how evaporation is treated in our

formulations.

To start, the use of the grey atmosphere approximation for the downwelling flux of longwave radiation is an approximation. It represents a more mechanistic approach of parameterising longwave radiative transfer, with the main difference to earlier



work (Kleidon and Renner, 2013a, b) being the additive constant $R_{l,0}$ in Eq. 5 that played here an important role. The use of the grey atmosphere approximation, however, is likely to overestimate the downward longwave flux for a given optical depth. Turbulent fluxes cause a lower surface temperature than in radiative equilibrium, which results in a colder lower atmosphere that is in radiative-convective equilibrium. This, in turn, should be associated with a lower downwelling flux of longwave

radiation. By using the grey atmosphere approximation, we do not account for this effect, which is likely to result in some biases in our formulation. This is likely to result in an overestimation of the sensitivity of surface temperature to changes in the optical depth. However, as we do not calculate optical depths or use observations, but rather adjust it to represent the global mean temperature or a given temperature change, the effect of this bias in the radiation parameterisation is likely to be small for our results.

We also did not consider specifically absorption of solar radiation within the atmosphere (which can be seen by comparing $R_{s,toa,avg}$ to $R_{s,avg}$ in Table 2). This atmospheric absorption would result in some diurnal variation of heat storage within the atmosphere over oceans. However, since our expressions do not explicitly depend on changes in atmospheric heat storage, the effect of this should not change our results.

   Another limitation relates to our representation of turbulent fluxes. We used the Carnot limit and the assumption that turbulent

fluxes operate near the limit of maximised power. Yet, the diurnal variation of heat storage in the lower atmosphere over land may need to be accounted for in the derivation of thermodynamic limits, which may then result in a different partitioning of energy fluxes at the surface. However, as long as the turbulent fluxes on land are proportional to the instantaneous value of absorbed solar radiation at the surface (which is a good assumption as turbulent fluxes on land show a strong diurnal variation), turbulent fluxes must then be small at night. The basic reasoning would then still apply that nighttime temperatures are more

sensitive to a change in greenhouse forcing, thus resulting in an altered climate sensitivity of land surfaces compared to ocean surfaces, although the specific ratio $\phi$ may differ from our value of 3/2.

   Furthermore, we do not explicitly consider evaporation in our formulation, but include it in the turbulent fluxes $J$. Evaporation and the associated latent heat flux cools the surface, yet it only heats the atmosphere (and the surface) when it condenses. At the global scale in steady state, evaporation needs to balance precipitation, so that evaporation does not necessarily need

to be represented as a separate term in the surface energy balance. Yet, spatiotemporal imbalances between evaporation and precipitation due to storage changes of water vapour and moisture transport can result in regional temperature variations due to evaporation (Kleidon and Renner, 2013a). Furthermore, differences in radiative parameters during dry and wet periods may result in further modulations of surface temperatures that we did not account for and that could have an effect (Rochetin et al., 2014). Those effects would clearly need to be addressed when our formulations are applied to the regional scale, which could

form a topic of future research.

   Yet, overall, it would seem that despite these deficiencies, our simple representation is able to adequately illustrate our explanation from the introduction in a parsimonious way as it captures the difference in climate sensitivity of ocean and land surfaces, and connects this difference to the difference in sensitivity between minimum and maximum temperatures.



## 4 Conclusions

We attributed the different climate sensitivities of ocean and land surfaces to the different way by which the surface-atmosphere system buffers the strong diurnal variations in solar radiation. This explanation was illustrated with a physically-based representation of the surface energy balance in which turbulent fluxes were constrained by thermodynamics and where the two

different means of buffering diurnal variations were incorporated. We then showed that our representation predicts a ratio of climate sensitivities of ocean and land surfaces that is very close to CMIP 5 simulations. We furthermore showed that our interpretation also predicts a difference in the sensitivity of minimum and maximum temperatures over land that is consistent with observations. We thus conclude that the difference in climate sensitivities is primarily due to the different means by which the diurnal cycle is buffered.

We can draw a few implications from this explanation. First, our results show that the diurnal dynamics of the surface energy balance of ocean and land surfaces are distinctively different. For the ocean, variations in solar radiation are buffered by heat storage changes below the surface, so that turbulent fluxes do not show much of a diurnal variation. As variations in solar radiation are buffered in the lower atmosphere over land, this symbolises the strong coupling between the land surface and the lower atmosphere, with this coupling nevertheless being constrained by the energy balance over day and night over

the whole surface-atmosphere system. Second, the success of this rather simple, yet physically-based representation of the surface-atmosphere system in capturing this basic difference in climate sensitivity suggests that more can be learned from such an approach in identifying the dominant controls on surface temperatures and the diurnal temperature range. In future work, this approach could be extended to understand regional variations in surface temperatures and the diurnal temperature range, how it relates to hydrologic cycling, and how it relates to different effects of solar versus greenhouse forcing. It would allow

for a parsimonious approach to better understand land surface-atmosphere interactions. Last, but not least, our explanation should also be applicable to the different climate sensitivity of the seasonal cycle. It is well known that winter temperatures increase more strongly than summer temperatures with global warming. Our interpretation here would suggest that this can be explained by winter conditions being mostly shaped by radiative exchange because heat advection would make the lower atmosphere stable, thus making surface temperatures more sensitive to changes in greenhouse forcing. These aspects would

seem to provide ample opportunity to extend this research in the future.

*Acknowledgements.* This research contributes to the "Catchments As Organized Systems (CAOS)" research group (FOR 1598) funded by the German Science Foundation (DFG). We acknowledge the World Climate Research Programme's Working Group on Coupled Modelling, which is responsible for CMIP, and we thank the climate modeling groups (listed in Table A1 of this paper) for producing and making available their model output. For CMIP the U.S. Department of Energy's Program for Climate Model Diagnosis and Intercomparison provides

coordinating support and led development of software infrastructure in partnership with the Global Organization for Earth System Science Portals.

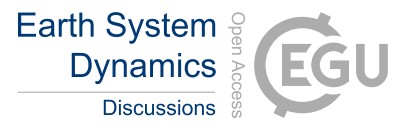

**Table A1.** Overview of CMIP5 model output data used in the study. Version denotes the used version of the dataset as provided by the CMIP5 data portal. Acronyms of CMIP5 models are taken from http://cmip-pcmdi.llnl.gov/cmip5/docs/CMIP5_modeling_groups.pdf.

| Model | Center | Period 4xCO2 | Period PI | Version 4xCO2 | Version PI |
|---|---|---|---|---|---|
| ACCESS1-0 | Commonwealth Scientific and Industrial Research Organization and Bureau of Meteorology, Australia | 400-449 | 750-799 | v2 | v20120329 |
| ACCESS1-3 | | 351-400 | 700-749 | v1 | v1 |
| bcc-csm1-1 | Beijing Climate Center, China Meteorological Administration | 260-309 | 451-500 | v1 | v1 |
| bcc-csm1-1-m | | 340-389 | 351-400 | v20120910 | v20120705 |
| CanESM2 | Canadian Centre for Climate Modelling and Analysis | 1950-1999 | 2961-3010 | v20111027 | v20120623 |
| CCSM4 | National Center for Atmospheric Research | 1951-2000 | 1251-1300 | v20120604 | v20130510 |
| CNRM-CM5 | Centre National de Recherches Meteorologiques / 1950-1999 Centre Europeen de Recherche et Formation Avancee en Calcul Scientifique | 2650-2699 | v20110701 | v20110701 | |
| CNRM-CM5-2 | | 1940-1989 | 2159-2208 | v20130402 | v20130402 |
| GFDL-CM3 | NOAA Geophysical Fluid Dynamics Laboratory | 96-145 | 451-500 | v20120227 | v20120227 |
| GFDL-ESM2G | | 246-295 | 446-495 | v20120830 | v20120830 |
| GFDL-ESM2M | | 251-300 | 446-495 | v20130214 | v20130214 |
| GISS-E2-H | NASA Goddard Institute for Space Studies | 1951-2000 | 2900-2949 | v20160505 | v20160511 |
| GISS-E2-R | | 1951-2000 | 4481-4530 | v20160505 | v20160511 |
| HadGEM2-ES | Met Office Hadley Centre (additional HadGEM2-ES realizations contributed by Instituto Nacional de Pesquisas Espaciais) | 1960-2009 | 2360-2409 | v20111129 | v20130114 |
| inmcm4 | Institute for Numerical Mathematics | 2190-2239 | 2300-2349 | v20130207 | v20130207 |
| IPSL-CM5A-LR | Institut Pierre-Simon Laplace | 2060-2109 | 2750-2799 | v20130506 | v20130506 |
| IPSL-CM5A-MR | | 1940-1989 | 2050-2099 | v20120114 | v20111119 |
| IPSL-CM5B-LR | | 1960-2009 | 2080-2129 | v20120430 | v20120114 |
| MIROC-ESM | Japan Agency for Marine-Earth Science and Technology, Atmosphere and Ocean Research Institute (The University of Tokyo) and National Institute for Environmental Studies | 101-150 | 2380-2429 | v20120710 | v20120710 |
| MIROC5 | Atmosphere and Ocean Research Institute, National Institute for Environmental Studies, and Japan Agency for Marine-Earth Science and Technology | 2201-2250 | 2620-2669 | v20120710 | v20120710 |
| MPI-ESM-LR | Max Planck Institute for Meteorology | 1950-1999 | 2800-2849 | v20120602 | v20120602 |
| MPI-ESM-MR | | 1950-1999 | 2800-2849 | v20120602 | v20120602 |
| MPI-ESM-P | | 1950-1999 | 2956-3005 | v20120602 | v20120602 |
| MRI-CGCM3 | Meteorological Research Institute | 1951-2000 | 2301-2350 | v20120701 | v20120701 |
| NorESM1-M | Norwegian Climate Centre | 101-150 | 1151-1200 | v20120412 | v20120412 |



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
