# Peer review of "An explanation for the different climate sensitivities of land and ocean surfaces based on the diurnal cycle"

_Earth System Dynamics, 2017_

## Referee Comment (RC1) · Anonymous Referee #1 · 30 May 2017

"An explanation for the different climate sensitivities of land and ocean surfaces based on the diurnal cycle" by Kleidon and Renner introduces a simple conceptual model to understand why surface temperatures over land respond more strongly than those over ocean to climate change. The mechanism is based on differences in the diurnal energy budgets over land and ocean and is found to agree well with predicted land-ocean warming contrasts from CMIP5 models.

I find this paper to be a novel and interesting addition to the literature on the land-ocean warming contrast. My main comments (outlined below) relate to the formulation of the simple model and the validity of these assumptions. If the authors can address these concerns, I will be delighted to recommend publications.

[Figure]

Specific comments: -Page 1, Line 2: "...with the cause for this difference being still unclear." I do not agree with this statement: Much research on the land-ocean contrast has been conducted over the last 10 years and in particular, the convective quasi-equilibrium theory by Byrne & O'Gorman (2013) [cited in this text] can quantitatively capture the warming contrast in CMIP5 models. So I, and many others in the field, strongly believe that we now do have a good understanding of the processes driving the warming contrast and I suggest that the authors might refine this statement in the abstract to reflect this developing consensus. -Page 1, Line 20: Spelling: "be found" -> "been found" -Page 1, Line 23: Byrne & O'Gorman (2013) identified that the land-ocean warming contrast depends not only on changes in relative humidity, but also on the climatological relative humidity over land. -Figure 1: "loosing heat" -> "losing heat"

-Pages 2,3: The following passage of text contains many statements that are key assumptions for the simple model derived in this study, yet are not supported by references. I strongly recommend that the authors better justify these statements by citing observational (preferably) or modeling studies:

"For ocean surfaces, these heat storage changes take place in the surface ocean. Solar radiation penetrates the surface ocean to quite some depth before it is absorbed. Combined with the large heat capacity of water, this results in diurnal heat storage changes that take place below the ocean surface (sketched by the red line on the left of Fig. 1). The build-up of heat storage during the day then maintains radiative cooling and turbulent heat fluxes during the night, resulting in little diurnal variations in surface temperature and turbulent fluxes. Over land this situation is quite different. Solar radiation is absorbed at the surface (or above in a canopy), but not below the surface. This is because land surfaces are not transparent as water, and because the heat conductivity of soils is generally so low that heat fluxes in and out of the ground are generally small. The strong diurnal variation in solar radiation is thus not buffered below, but rather above the surface in the lower atmosphere. These changes in heat storage manifest themselves in the diurnal growth of the convective boundary layer.
This buffering above the surface has an important consequence for the fluxes of the surface energy balance. Turbulent fluxes only take place when the surface is heated by solar radiation during the day that causes the near-surface air to become unstable, while the nighttime is characterised by stable conditions near the surface as little heat can be drawn from the heat storage below the surface. This prevents turbulent fluxes to take place at night and the cooling at night is thus determined only by radiative exchange. Turbulent cooling of the surface thus takes place during half of the whole day, while the other half it is cooled by radiative exchange."

-Page 5, line 7: "which is typically small on a diurnal time scale" -> reference needed to support this statement -Page 7, line 14: "For the land surface, we assume that the heat storage changes take place in the lower atmosphere" -> is this a reasonable assumption? Reference needed again. Land surfaces can get very hot during the daytime so it is not obvious to me that the surface storage term should be negligible. -Page 7, line 20: Is it reasonable to assume a net LW radiative flux of zero overnight over land? Are there observations to support this key assumption? -Page 8, section 3.1 & Table 2: Various numbers are assigned to parameters in the simple model here but it is not clear which are based on observations and which are tuned so as for the simple model to give a reasonable climatology. A but more detail behind these choices is requested. -Page 8, line 26: The change in LW optical depth is chosen to be 0.11 - how does this compare to observed/modeled radiative forcings? It is important that this number is reasonable as this would validate the simplified radiation parametrisation used. -Page 13, lines 10+11: " First, our results show that the diurnal dynamics of the surface energy balance of ocean and land surfaces are distinctively different." -> I would argue that you assume the diurnal dynamics are different when constructing the simple model. More justification for these assumptions is greatly needed in order to make the results more compelling.

---

## Referee Comment (RC2) · Anonymous Referee #2 · 10 Jul 2017

Review of paper "An explanation for the different climate sensitivities of land and ocean surfaces based on the diurnal cycle" by Kleidon and Renner.

First please accept my apologies for taking over a month to return this review.

This paper is related to a key feature of the Earth system, which is noted in both measurement record and in Earth System Model projections. That is under increasing atmospheric greenhouse gas concentrations, the land surface is in general warming faster than that of the oceans. This has important implications for society. For instance, any stabilisation target (such as global warming capped to two degrees) will be related to higher final temperatures over land, and this needs to be planned for. A sentence to

this effect could be added to the Conclusions maybe?

Despite the land-sea warming contrast being one of the few robust findings between GCMs, there has been relatively little explanation of this to date. Here, a fascinating process description is given, based on differential abilities of the land and ocean to distribute heat over the diurnal cycle. In particular, the lack of turbulent transfer above the land surface makes this more sensitive to radiation changes, adjusted by rising GHG concentrations.

Possibly the only request for additional work is as follows. If this is possible (as no wish to delay the publication) - would it be possible to solve the equation set and present daily profiles? In other words, a diagram with x-axis as hours 0-24, along with a y-axis that plots dUs/dt. Could this be achieved in 4 cases? Pre-industrial for land and ocean, and (say) 4xCO2 also for land and ocean? This would then compliment what I assume are stylised lines, for dUs/dt in Figure 1.

In terms of technical description and completeness, then a couple more lines describing the equation closure might help? So around Section 2.3, p6 lines 10- p7 lines 3. This depends strongly on the "maximum power" assumption for closure. Details are in Kleidon and Renner 2013a, but at little more description might help here. Also, is there a more general theoretical physics reference from before 2013?

In many ways, this paper opens more questions than it answers. But that's not to complain, and simply an indication that it has strong potential to be cited. Some of these questions the authors could hint at in their Conclusions? (indeed they state "These aspects would seem to provide ample opportunity to extend this research in the future"). This could be:

(i) Is there any compatibility between the analysis here and other earlier work and explanations mentioned in Introduction?

(ii) Can the seasonal cycle be analysed in more detail – for instance, towards the poles,
there will be a large differentiation between winter and summer day length, which might be seen in the data? Presumably temperature differentiation would become bigger in summer months?

(iii) Can sub-daily data be analysed from ESMs to verify more the model presented here?

(iv) If the model description, concept and formulation is accurate, then could it be inverted to tell us (based on temperature measurements) better parameterisations of turbulent transport?

(v) Are the equations sufficiently simple that there might even be room for analytical solution, especially if conceptual descriptions were given to the solar drivers. These could be as the positive part of a sin wave, or a parabolic description for solar forcing during daytime hours.

The paper is to be applauded for providing a complete model, along with all variables defined with units (in Table 1). This allows other researchers to try and build the model components if needed.

In summary, there I think this paper should be published, and it contributes strongly to the discussion surrounding why both measurements and ESMs indicate differential and higher warming over land than oceans. There are a few points above if the authors were interested in making a new manuscript version, with minor revisions.

---

## Author Comment (AC2) · 15 Jul 2017

We thank the reviewer for the constructive and supportive comments made on the manuscript.

***Reviewer comment 1:*** *This paper is related to a key feature of the Earth system, which is noted in both measurement record and in Earth System Model projections. That is under increasing atmospheric greenhouse gas concentrations, the land surface is in general warming faster than that of the oceans. This has important implications for society. For instance, any stabilisation target (such as global warming capped to two degrees) will be related to higher final temperatures over land, and this needs to*

[Figure]

*be planned for. A sentence to this effect could be added to the Conclusions maybe?*

**Reply:** Yes, we will adjust the conclusions in the revision and point out the societal implications of the study.

*Reviewer comment 2: Possibly the only request for additional work is as follows. If this is possible (as no wish to delay the publication) - would it be possible to solve the equation set and present daily profiles? In other words, a diagram with x-axis as hours 0-24, along with a y-axis that plots dUs/dt. Could this be achieved in 4 cases? Pre-industrial for land and ocean, and (say) 4xCO2 also for land and ocean? This would then compliment what I assume are stylised lines, for dUs/dt in Figure 1.*

**Reply:** Thanks, this is a very good suggestion! This figure can easily be done. We will include such a figure in the revision.

*Reviewer comment 3: In terms of technical description and completeness, then a couple more lines describing the equation closure might help? So around Section 2.3, p6 lines 10- p7 lines 3. This depends strongly on the "maximum power" assumption for closure. Details are in Kleidon and Renner 2013a, but at little more description might help here. Also, is there a more general theoretical physics reference from before 2013?*

**Reply:** We will provide more details as suggested.

*Reviewer comment 4: In many ways, this paper opens more questions than it answers. But that's not to complain, and simply an indication that it has strong potential to be cited. Some of these questions the authors could hint at in their Conclusions? (indeed they state "These aspects would seem to provide ample opportunity to extend this research in the future"). This could be: (i) Is there any compatibility between the analysis here and other earlier work and explanations mentioned in Introduction? (ii) Can the seasonal cycle be analysed in more detail – for instance, towards the poles, there will be a large differentiation between winter and summer day length, which might*

*be seen in the data? Presumably temperature differentiation would become bigger in summer months? (iii) Can sub-daily data be analysed from ESMs to verify more the model presented here? (iv) If the model description, concept and formulation is accurate, then could it be inverted to tell us (based on temperature measurements) better parameterisations of turbulent transport? (v) Are the equations sufficiently simple that there might even be room for analytical solution, especially if conceptual descriptions were given to the solar drivers. These could be as the positive part of a sin wave, or a parabolic description for solar forcing during daytime hours.*

**Reply:** These are excellent suggestions! We will include them in the discussion/conclusion at the end of the manuscript.

*Reviewer comment 5: In summary, there I think this paper should be published, and it contributes strongly to the discussion surrounding why both measurements and ESMs indicate differential and higher warming over land than oceans. There are a few points above if the authors were interested in making a new manuscript version, with minor revisions.*

**Reply:** Thank you very much for the helpful suggestions!

―――――――――――――――――――

---

## Author Response (AR1)

**Response to Reviews of "An explanation for the different climate sensitivities of land and ocean surfaces based on the diurnal cycle" by Kleidon and Renner**

Dear Editor,

We would like to submit our revised version of our manuscript. In this revision, we addressed the helpful and constructive points of the reviewers and the editor, and made minor modifications, including adding two figures for illustration. Hopefully, this has improved the manuscript so that it is acceptable for publication.

In the revised manuscript, we marked altered text by color, with blue for responding to reviewer 1, red for reviewer 2, and purple for the editor comments and other minor changes to the text for clarity. In the following, we put the comments in italic and our response in how we addressed these points in plain text. Some of these responses are based on the comments already posted in the discussion forum.

**1. Response to Review 1 (blue text in the revised manuscript)**

*1.1: "An explanation for the different climate sensitivities of land and ocean surfaces based on the diurnal cycle" by Kleidon and Renner introduces a simple conceptual model to understand why surface temperatures over land respond more strongly than those over ocean to climate change. The mechanism is based on differences in the diurnal energy budgets over land and ocean and is found to agree well with predicted land-ocean warming contrasts from CMIP5 models. I find this paper to be a novel and interesting addition to the literature on the land-ocean warming contrast. My main comments (outlined below) relate to the formulation of the simple model and the validity of these assumptions. If the authors can address these concerns, I will be delighted to recommend publications.*

We hope that the revision of the manuscript satisfactorily addressed the following concerns.

*1.2: Page 1, Line 2: "...with the cause for this difference being still unclear." I do not agree with this statement: Much research on the land-ocean contrast has been conducted over the last 10 years and in particular, the convective quasi- equilibrium theory by Byrne & O'Gorman (2013) [cited in this text] can quantitatively capture the warming contrast in CMIP5 models. So I, and many others in the field, strongly believe that we now do have a good understanding of the processes driving the warming contrast and I suggest that the authors might refine this statement in the abstract to reflect this developing consensus.*

We have removed this sentence from the abstract.

*1.3: Page 1, Line 20: Spelling: "be found" -> "been found"*

This has been corrected in the revision.

*1.4: Page 1, Line 23: Byrne & O'Gorman (2013) identified that the land- ocean warming contrast depends not only on changes in relative humidity, but also on the climatological relative humidity over land.*

The text was adjusted in the revision.

*1.5: Figure 1: "loosing heat" -> "losing heat"*

The figure was corrected.

*1.6: Pages 2,3: The following passage of text contains many statements that are key assumptions for the simple model derived in this study, yet are not supported by references. I strongly*

*recommend that the authors better justify these statements by citing observational (preferably) or modeling studies.*

We have added references to these statements and rewritten parts of the paragraph to more clearly distinguish which parts are well established and which parts are our interpretation of these observations in the revised manuscript.

*1.7: Page 5, line 7: "which is typically small on a diurnal time scale" -> reference needed to support this statement.*

References have been added to the text.

*1.8: Page 7, line 14: "For the land surface, we assume that the heat storage changes take place in the lower atmosphere" -> is this a reasonable assumption? Reference needed again. Land surfaces can get very hot during the daytime so it is not obvious to me that the surface storage term should be negligible.*

References have been added to the introduction, where the magnitude of the ground heat flux is first discussed.

*1.9: Page 7, line 20: Is it reasonable to assume a net LW radiative flux of zero overnight over land? Are there observations to support this key assumption?*

We have added a reference to the textbook by Oke (1987) about this general feature.

*1.10: Page 8, section 3.1 & Table 2: Various numbers are assigned to parameters in the simple model here but it is not clear which are based on observations and which are tuned so as for the simple model to give a reasonable climatology. A but more detail behind these choices is requested.*

We have modified the text to more clearly describe how the tuning was done, and added a new figure (Fig. 3) to better illustrate this.

*1.11: Page 8, line 26: The change in LW optical depth is chosen to be 0.11 - how does this compare to observed/modeled radiative forcings? It is important that this number is reasonable as this would validate the simplified radiation parametrisation used.*

An additional figure (Figure A1) is provided and text was added to section 3.1 to address this point. (Note that there was an error in the stated value of the change in LW optical depth. The actual difference is actually 0.18, see point 4.1 below.)

*1.12: Page 13, lines 10+11: "First, our results show that the diurnal dynamics of the surface energy balance of ocean and land surfaces are distinctively different." -> I would argue that you assume the diurnal dynamics are different when constructing the simple model. More justification for these assumptions is greatly needed in order to make the results more compelling.*

We hope that by the clarifications added during the revision (particularly in the introduction) we satisfactorily addressed this point.

**2. Response to Review 2 (red text in the revised manuscript)**

*2.1: This paper is related to a key feature of the Earth system, which is noted in both measurement record and in Earth System Model projections. That is under increasing atmospheric greenhouse gas concentrations, the land surface is in general warming faster than that of the oceans. This has important implications for society. For instance, any stabilisation target (such as global warming*

*capped to two degrees) will be related to higher final temperatures over land, and this needs to be planned for. A sentence to this effect could be added to the Conclusions maybe?*

We have reformulated the conclusions of the paper, focussing on this aspect. The broader discussion on potential future work etc. has been moved to the new section 4.3.

*2.2: Possibly the only request for additional work is as follows. If this is possible (as no wish to delay the publication) - would it be possible to solve the equation set and present daily profiles? In other words, a diagram with x-axis as hours 0-24, along with a y-axis that plots dUs/dt. Could this be achieved in 4 cases? Pre-industrial for land and ocean, and (say) 4xCO2 also for land and ocean? This would then compliment what I assume are stylised lines, for dUs/dt in Figure 1.*

We have followed the reviewers suggestion and included such a figure in the revised manuscript (new Fig. 4) as well as a paragraph to describe this figure in section 3.1.

*2.3: In terms of technical description and completeness, then a couple more lines describing the equation closure might help? So around Section 2.3, p6 lines 10- p7 lines 3. This depends strongly on the "maximum power" assumption for closure. Details are in Kleidon and Renner 2013a, but at little more description might help here. Also, is there a more general theoretical physics reference from before 2013?*

The description of the maximum power assumption has been extended in section 2.3.

*2.4: In many ways, this paper opens more questions than it answers. But that's not to complain, and simply an indication that it has strong potential to be cited. Some of these questions the authors could hint at in their Conclusions? (indeed they state "These aspects would seem to provide ample opportunity to extend this research in the future"). This could be:*
*(i) Is there any compatibility between the analysis here and other earlier work and explanations mentioned in Introduction?*
*(ii) Can the seasonal cycle be analysed in more detail – for instance, towards the poles, there will be a large differentiation between winter and summer day length, which might be seen in the data? Presumably temperature differentiation would become bigger in summer months?*
*(iii) Can sub-daily data be analysed from ESMs to verify more the model presented here?*
*(iv) If the model description, concept and formulation is accurate, then could it be inverted to tell us (based on temperature measurements) better parameterisations of turbulent transport?*
*(v) Are the equations sufficiently simple that there might even be room for analytical solution, especially if conceptual descriptions were given to the solar drivers. These could be as the positive part of a sin wave, or a parabolic description for solar forcing during daytime hours.*

In the revision, we extended the discussion on these topics in the new sections 4.2 and 4.3.

**3. Response to Editor's comments (purple text in the revised manuscript)**

*3.1: Regarding the first several comments of referee 1, I encourage the authors to look for observational literature references that can be added to the textbook references. (Just because something is in a text book doesn't necessarily mean it is actually correct!) While I don't seriously doubt that these assertions are, indeed, correct, since these points are central to the arguments in this paper, having more direct observational references would strengthen the paper. (And also allow readers to access the magnitude of deviations from the assumptions made.)*

In the revision, some review/synthesis papers were added along with references to textbooks, specifically on the issue of the diurnal cycle over oceans and the role of the ground heat flux.

*3.2: Please make sure the references to the textbooks that are used are given in proper format in the revised manuscript.*

Yes, proper references to the textbooks were given in the revised manuscript. Also, we added some review/synthesis papers regarding the diurnal variation over ocean and regarding the ground heat flux on land. Regarding the diurnal variation of the surface energy balance over land, this is so well and so repeatedly observed that (to our knowledge) there is no recent review paper that one could cite, so we believe that the citation to textbooks is appropriate.

*3.3: For the reply "Observations of the land surface energy balance generally show a net longwave cooling at night with values typically well below 100 W m−2. " it would be useful to provide an observational reference for this. (or, if this is not readily available, show this information in supplementary materials.)*

A reference was added to the textbook of Oke (1987) (a classic textbook on boundary layer/land surface climatology). It is such a common observation over land that it would seem inappropriate to select one specific paper that presents measurements of a single isolated site to make reference to.

*3.4: In the response to Reviewer 1 comment 11:, please add this analysis, associated discussion, and related figure wither in the revised manuscript or, perhaps, as either supplementary material or an appendix in order to not breakup the flow of the manuscript.*

The material was added in the Appendix as Figure A1.

*3.5: In line with some of the reviewer comments, I urge the authors in their revision to discuss a bit more how the current analysis could be extended. Again, the referee's have agreed that the author's arguments are sound - so the purpose of the paper should not solely be to justify the methodology and assumptions used in the current analysis. It would improve the contribution of this paper to the literature to discuss a bit more the potential implications of taking into consideration the next order of analysis. (For example, could additions improve the comparison between this analysis and the CMIP models in Figure 3? In particular I note that this figure seems to show that the result from this analysis appears to be at the low end of the model range. Given the author's methodology and interpretation, what inter-model differences might explain this range?)*

To provide more space for discussing this aspect, we added a separate discussion section (section 4) in the revision and shortened the conclusions accordingly. This new discussion section includes the limitations (section 4.1), the relationship to previous explanations of the different climate sensitivity (section 4.2), and potential for further research.

We also address the point why our estimate seems to describe the low end of the model range. The main reason for this effect is likely that the same reasoning regarding stability also applies to wintertime conditions. That is, winter in high latitude regions is also dominated by stable conditions due to the lack of solar radiative heating, thus making them also more sensitive to radiative change. That high latitudes are more sensitive to greenhouse forcing is very well known and documented, and could thus be explained by the same mechanism. We point out in section 4.3 that this would be an interesting extension of this work.

*3.6: In the discussions of the current simplifying assumptions it would be useful if the authors could also discuss (perhaps at the end of the paper) the potential implications if these analysis were taken a step further. For example, in the revised introduction the authors note that "the ground heat flux does typically not exceed more than 100 W m−2, which is comparatively small to the absorption of 800 W m−2". What might be the implications of taking this into account? Would this be a O(10%) correction to the result?*

This is an interesting aspect. The direct effect of this is not so easy to estimate, because regions in which this heat flux is large (e.g., deserts) are also typically regions of low optical thickness and

thus have a different radiative forcing than the global mean, as considered in this paper. We included some discussion in the revised manuscript on this aspect in the new section 4.3.

*3.7: How about the opposite case, moist land-surface regions (including rivers, wetlands, lakes, etc.)? Presumably there would be greater diurnal heat buffering there as compared to a desert area? Would incorporating such differences significantly alter the estimate?*

We have to some extent included these items in the new section 4.3.

**4. Other changes during revision (purple text in the revised manuscript)**

4.1: There was an error in the text regarding the change of the optical depth ($\Delta tau = 0.18$, not 0.11). The text as well as the resulting value for $\Delta R_{l,d}$ was adjusted. Other results were not affected by this error.

4.2: We added a bar diagram (new Fig. 3) to better illustrate the energy balance partitioning in the global mean using the values from Table 2.

4.3.: The illustration of the sensitivity to solar radiation was slightly modified to a scenario of solar brightening (section 3.3). This was done because after discussions with colleagues, this setup is more plausible and easier to communicate. It does not, however, qualitatively affect the results or conclusions.

4.4: To better illustrate the different effects of changes in the greenhouse forcing compared to absorption of solar radiation, a new Fig. 5 was added to illustrate the values from Table 2.

4.5: Minor parts of the text was changed for clarity.

Kind regards,

Axel Kleidon
(on behalf of the authors)